# Meaningful everyday life situations from the perspective of children born preterm: A photo-elicitation interview study with six-year-old children

**Anna Karin Andersson** [1] *, **Lena Almqvist** [1], **Katarina Strand Brodd** [2], **Maria Harder** [1]

1 School of Health, Care and Welfare, Mälardalen University, Västerås, Sweden, 2 Department of Women´s and Children's Health, Perinatal, Neonatal and Pediatric Cardiology Research Uppsala University, Uppsala, Sweden

* anna.karin.andersson@mdu.se

## Abstract

### Aim

The aim of the study was to explore meaningful everyday life situations as perceived by six-year-old children born preterm.

### Materials and methods

The study had a descriptive qualitative design with an inductive approach. Ten, six-year-old children born preterm, not diagnosed with any disabilities, participated. Data was collected by photo-elicitation interviews to stimulate and help the children to describe their meaningful everyday life situations. A qualitative content analysis according to Elo and Kyngäs was applied.

### Results

The children's descriptions of meaningful everyday life situations can be understood as *being in an active and dynamic process*, representing the core category. The analysis resulted in three generic categories, as the children described the significance of *having significant circumstances* and *doing things*. The experiences the children gain when they do things create their *desire for further development*.

### Discussion

The results reveal that children born preterm are able to reflect on and give detailed descriptions of situations of importance to them. The study suggests that if six-year-old children born preterm are given the opportunity to share their views they can take an active role e.g. in planning and carrying through of interventions by health care services.

**Data Availability Statement:** All relevant data are within the paper and its Supporting information files. Of confidentiality reasons, photos taken by

the participants and recorded interviews will not be available.

**Funding:** The author(s) received no specific funding for this work.

**Competing interests:** The authors have declared that no competing interests exist.

## Background

Meaningful everyday life situations contribute to children's development as they occur in their immediate environment where they gain experiences, have an active role, and interact socially [1, 2]. Family dinner, play time, and bedtime are examples of these everyday life situations [3, 4] and commonly, typically developed children participate in a diversity of everyday activities [5]. Children's everyday life situations may be influenced by circumstances such as being born preterm.

Research shows that children born preterm can have difficulties in their everyday life situations [6–11]. These difficulties concern self-care routines, mobility [7, 11], play, leisure, and social interaction [8, 9, 11]. However, everyday life situations for children born preterm have mainly been explored from the parents' perspective, i.e. the parents act as proxies for their child [6–11] and this does not clarify what the difficulties mean for the children themselves or what they perceive as meaningful everyday life situations. The children have the right to be heard in matters that concern them [12] and the evidence of children being trustworthy as informants in matters concerning their everyday life [13, 14] also embraces children born preterm.

Children's perceptions of meaningful everyday life situations may be multidimensional and related to various factors such as personality, development, and context. Also, meaningful everyday life situations can be linked to the child's preferences i.e opportunity to choose and to be able to take part in activities. Preferences are described, along with activity competence, in the model of Family of Person-related Constructs (fPRC), as intrinsic person-related concepts that are related to children's participation [15]. The fPRC describes the influence of the intrinsic person-related constructs for past and present participation as influential for future participation outcomes.

Furthermore, children's perceptions of meaningful everyday life situations may differ from their parents' perceptions on the same issue [16]. Rosenberg and Bart, who use the concept of 'enjoyment' rather than meaningfulness, found that children's enjoyment in everyday life situations was related to their emotional functioning, while their parents related child enjoyment to skills and age [16]. Children's perceptions of meaningful everyday life situations can be related to their participation in the everyday life situations they prefer or are interested in [15]. This implies that knowledge of children's perceptions of what is meaningful in their everyday life is acquired by asking the children themselves [17], i.e., by taking the child's perspective [18]). Children aged four to five have described how having skills in various activities, being able to play a lot and to play with friends are important to them [19]). Furthermore, children aged five to six have described the importance of spending time in the various preschool settings such as playgrounds, the kitchen, and the dressing room where they put on and take off clothes after outdoor activities. These children also describe other children as being their friends and say that having access to specific playthings, e.g., blocks, is important for them in preschool [20]. The knowledge about what is meaningful for children in their everyday life situation does not automatically involve children born preterm. Although most children born preterm do not develop disabilities [21], they may have experiences and difficulties that can affect their development and participation in everyday life situations [22]. Children born preterm are considered a risk group leading to attention and care by health services through childhood to a higher extent than term-born children. Additionally, the experience of a perhaps life-threatening start can influence parenthood. The vulnerability of the preterm baby, as perceived by the parents, may linger through childhood, and manifest a conception of the child as a vulnerable individual. The aim of the study was to explore meaningful everyday life situations as perceived by six-year-old children born preterm.

## Materials and methods

### Design

A descriptive qualitative design with an inductive approach was applied.

### Ethical considerations

To include children in research requires particular considerations. In the present study, the children and their parents were informed verbally and in writing about the study. Written informed consent from the parents was obtained. The children were first asked for their consent to participate by their parents, a person they knew well. Verbal consent was obtained from the children as the question about participating was also posed as an introduction to the interview. Both child and parent were free to withdraw from the study without any requirement to state a reason. As the focus was on the child's perspective with respect to the perceptions of the child, the risk of feeling singled out when participating and taking photographs may have been minimised. The study, including the information letter to participants and the consent letter, was approved by the Regional Ethical Review Board in Uppsala (Dnr 2016/240).

### Participants

With support from the medical secretary and the neonatologist at the paediatric clinic in a middle-sized county council in Sweden, five to six-year-old children born preterm (n = 119), who had been cared for at the neonatal ward, were invited to take part in the study. Ten, six-year-old children born preterm, five boys and five girls, chose to participate. The gestational age at birth for these children ranged from 26–36 weeks, with one child born in week 26 and nine between weeks 32–36, and they had not been diagnosed with neurological, hearing, visual and/or intellectual disabilities (Table 1). The reason for declining participation is not entirely known; some parents stated lack of time and one child declined participation for unknown reasons.

### Data collection

Data was collected by photo-elicitation interviews [20, 23]. To promote the children's involvement in the data collection and to elicit their perceptions of meaningful everyday life situations, the children were asked to take photographs of meaningful everyday life situations

**Table 1. Description of children's characteristics.**

| Child | Age | Gestational age at birth | Accompanying care-giver | Location of photographs |
|---|---|---|---|---|
| Boy | 6 | 35 | Mother | Home, outside, sports arena |
| Boy | 6 | 36 | Mother | Home |
| Boy | 6 | 26 | Mother | Home, outside, school |
| Boy | 6 | 36 | Father | Home, school, sports arena |
| Boy | 6 | 36 | Mother and father | Home |
| Girl | 6 | 34 | Mother | Home, outside, sports arena |
| Girl | 6 | 35 | Father | School |
| Girl | 6 | 33 | Mother and father | Home, outside, school, sports arena |
| Girl | 6 | 35 | Mother | Home |
| Boy | 6 | 35 | Father | Home, outside, school |
| Girl | 6 | 32 | Mother and father | Home, school, stable |

[17, 20]. The children's photographs were then used in the interviews to stimulate and help the children to describe their meaningful everyday life situations. The photos were not analysed or interpreted separately.

## Photographing

The children were offered the opportunity to borrow a digital camera from the research group. Four children and their parents chose to use their own camera as it was familiar to the child. The children were asked to take as many photographs as they liked of their everyday life situations in their immediate environment during a period of two weeks. If children wanted to take photographs at school, the principal and the teacher were contacted by the first author to receive permission for the child to take photographs during school hours. The children's parents and teachers were asked to assist the child during the photo-shooting but without influencing the child's choice of everyday life situations. The children's photographs included pictures taken at home, school and during leisure activities, indoors and outdoors, and they illustrated the children themselves, other persons and objects. Before the individual interviews the children were requested to select a maximum of ten photos, and the parents were asked not to influence the child's choice. Despite the instruction to bring a maximum of 10 photographs, the number of photographs each child selected ranged from 8–49. Altogether, 110 photographs were selected to show and talk about in the interviews (Table 2).

## Interviewing

The individual interviews took place at a location preferred by the children and their parents: at home, at the university or at the paediatric clinic. During the interview, the children were asked to talk about the situations depicted in the photographs: why the situation had been picked out; the context of the situation; the objects and/or persons displayed; how often the situation occurred; and how the child felt in, and about this situation. The interviews were conducted by the first author who was a physiotherapist with experience in consulting with children of preschool ages. The interviewer had no prior clinical or other relation to the children or their parents. In the interview session the children were accompanied by one or two parents who were asked to remain silent. With permission from the parents and children, the interviews were recorded by the first author. One child did not consent to the recording and notes were taken during the interview.

**Table 2. What the children photographed using a digital camera.**

| Object | Number of photographs (N = 128) |
|---|---|
| Play: Building Lego or other, relation-play, other | 22 |
| Persons: parents, siblings, friends grandparents, friends | 17 |
| Games: Computer- or TV-games, tabletop, card games | 13 |
| Arts and Crafts: Drawing, cutting, other | 12 |
| Sports: Formal and informal outside | 19 |
| Household activities: Assist in cooking, do the dishes | 6 |
| Reading | 5 |
| Learning: Attend school, learn new things | 8 |
| Be self | 2 |
| Special events: Christmas, Halloween, | 3 |

## Data analysis

A qualitative content analysis according to Elo and Kyngäs was performed and triangulation of the data analysis was performed in a joint process between the first and second author [24]. An overview of the data analysis process is presented in Table 3. The authors (AKA, MH) separately highlighted the meaning units representing the study aim for comparisons of conformity between the authors. When agreement was reached on the meaning units, they were coded separately by the authors in relation to the aim. The codes were discussed going back and forth between the transcribed text and the constructed codes until agreement was reached. The codes were scrutinised for similarities and differences in a joint process between the authors (AKA, MH), and then the codes belonging together were grouped together as subcategories (Table 3). The subcategories were subjected to the same scrutiny and those belonging together were constructed as generic categories (AKA, MH) indicating that the children perceived meaningful everyday life situations as: doing things; having the significant circumstances needed to do things and having the desire for further development. During the analysis there was an attempt to maintain the child's perspective by analysing what was actually said and not abstracting the categories too far. The photographs were used in the analysis to increase the understanding of the children's descriptions.

# Results

The children's descriptions of meaningful everyday life situations can be understood as *being in an active and dynamic process*, representing the core category. The analysis resulted in three generic categories, as the children described the significance of *having significant circumstances* and *doing things*. The experiences the children gain when they do things create their *desire for further development*. The categories are presented in Table 4.

## To have significant circumstances

The children's significant circumstances include their current stock of significant others, skills and things/objects needed to do things. These circumstances are essential to and necessary for the children's everyday life and are described as something they like. The children say that their significant others are the persons and animals involved in their everyday life. The significant persons are parents, siblings, grandparents and friends. The significant animals are dogs,

**Table 3. Overview of the data analysis process.**

| Meaning units | Code | Subcategory | Generic category | Core category |
|---|---|---|---|---|
| Having a girlfriend who climbs [on a climbing wall] to the top and who dances ballet.<br><br>Being with mom means doing a lot of things together and she comforts me when I'm sad, she takes care of me and sets boundaries, because I love her, she's like me and she loves me | To have parents, a family, relatives, girlfriends, pets who knows different things like comforting, setting boundaries, taking care of me | To have significant others | Significant circumstances to do things | Being in an active and dynamic process |
| To build machines and spaceships of LEGO<br><br>To read books like Nelly Rapp and Lasse-Maja yourself. It is good to be able to read. To train your voice [when reading] and it's fun with the bookshelves | To do fun things like building with LEGO, reading books | To do things that are fun or important | Doing things | |
| To dream of going to the climbing club<br><br>Wanting to work with animals when I grow up. When my little brother and I grow up, we will work on a farm with different animals | Wishing to be able to go to the climbing club, work with animals | To have desire for significant activities | Desire for significant development | |

**Table 4. Overview of the results including subcategories, generic categories and core category.**

| Subcategories | Generic categories | Core category |
|---|---|---|
| To have significant others To have significant skills To have significant things | Significant circumstances to do things | Being in an active and dynamic process |
| To do things that are fun or important To do things with significant others To do things in significant places or contexts | Doing things | |
| To have desire for significant others To have desire for significant skills To have desire for significant activities | Desire for significant development | |

cats and horses. The children describe the significant persons as nice as they give presents, sew costumes and construct furniture. They are also good at various activities such as climbing and dancing. Furthermore, these persons are important and valuable as they take care of the children by comforting them, giving advice, and setting limits which contribute to safety and mutual love. The significant animals are nice as they are obedient, beautiful, soft to stroke, and can do tricks.

'To be with my dog and stroke her and she follows my instructions: Lie down, go, sit, sit back, I like that. The dog follows my grandfather and me on our walks'

[b20]

The children say that their significant skills include being able to and being good at various things such as cutting, gluing, drawing, building boats and aeroplanes with Lego and preparing pancake batter. Significant skills also include swinging fast on the swings, having the courage to run and jump on a trampoline, climbing trees, horseback riding and skating forwards and backwards. The described skills also involve being able to master one's temperament and having knowledge about astronomy.

The children talk about significant things/objects such as fancy and funny clothes, costumes, raspberry bushes, a safety box to store valuable things such as Pokémon cards, and a bed to rest in and get new energy.

## Doing things

The children's descriptions of doing things involve *doing things that are fun or important*, *doing things with significant others* and *doing things in significant places or contexts*. *Doing things* is described as essential in the children's everyday life. There is a relation between doing things and significant others and between doing things and significant places as certain persons or places are connected to certain activities. External conditions including the children's significant others, their significant places or context shape the children's doing: free play as "mother-father-child" is shaped by the available objects e.g., furniture in a specific place/ location.

The children say that *doing things that are fun or important* are the things they enjoy doing, are good at and do often. These things have aspects of creating, collecting, entertaining, being physically active, and role-playing. The aspects of creating and collecting include building things with Lego, baking, preparing food, drawing, cutting, and doing arts and crafts. Furthermore, they include saving money for and collecting Pokémon cards and winning things. The aspects of entertaining include playing games such as cards or board games but also video and computer games, visiting museums, and going on excursions and journeys. Aspects of

entertaining also include winning things in a lottery; listening to fairy tales; watching TV; reading books or reading Pokémon comics. The children's descriptions of things with aspects of physical activity include taking walks, climbing on a climbing frame, climbing walls or trees, practising ice-hockey or Taekwondo, swimming, dancing, and going on snow racers. The role-playing aspect includes cooking play food, playing "mother-father-child", pretending to be spies, and dressing in costumes. The following quotation shows an aspect of creating:

'Making figures out of toilet paper rolls, wool balls, beads, green shapes of clothes is fun'

[b90]

Other aspects of doing things that are fun and important are hanging out and being affectionate, for example: having cosy moments, talking, hugging, caressing, and giving help.

The children say that *significant others are the ones with whom they do the things* that are fun and important. This may be understood to mean that doing something with a significant other makes the activity more fun and meaningful. The significant others include both people and pets. The significant persons in the children's everyday life are those they encounter and do things with daily: their family members e.g. parents and siblings; other children e.g. their best friends and school mates, or other adults such as their teachers or coaches in sports, e.g. Taekwondo. Other significant persons in the children's everyday life are the ones they encounter on a regular basis weekly or on specific occasions such as celebrations e.g. their relatives as grandparents or their sports coaches.

The children say that with friends and school mates they are physically active by e.g., climbing trees or practising ice-hockey, doing entertaining things as playing games, dancing, singing, or playing music. With siblings and parents, the children say they like doings things such as making food or cookies.

Doing things with the parents also includes being affectionate and having cosy moments. The children say that it is fun to be with the siblings and that they are available to play with. Furthermore, the children say that with their grandparents they enjoy doing things such as doing crafts, taking walks, visiting museums, and going on excursions and journeys. The children say they like playing with their pets, stroking them, and taking walks with them as the pets are kind and stay close. One child says:

'And it does not like to cuddle but [the dog likes] so much to play and if it detaches from the dog-leap it stays close'

[b160]

The children say that a *significant context* in their everyday life is where they do the things that are fun and important and such contexts are also related to their significant others. The contexts include their home, the family car, the grandparent's homes, the school, outdoors and the context of leisure activities. The children's description of their home context is more specific, for example including being in the kitchen making cakes and hanging out talking to family members or being in their own room doing entertaining things such as dressing up in costumes and watching films. In the school context the children describe the classroom, library, school gym, play corners and drama room as significant for doing entertaining things such as playing games, borrowing books to read and free playing. The children say that it is fun to be outdoors where they are physically active by riding bikes, playing on swings, and climbing trees. Furthermore, significant contexts are where they engage in leisure activities such as the stable for horseback riding courses, the climbing gym, or the municipal baths

where they learn swimming. One child recognises the family car as a significant context where to take a bumpy ride and sing and play:

'To ride the car with dad and go bumpy, laughing and joking, sitting and looking out the window, dancing, listening to music and playing Harry Potter, I think that is fun'

[b130]

### To have desires for development

The children's desires for significant development include their wish to expand their current stock of significant others, skills, and activities. These desires are related to their previous experiences in various everyday situations and can be expressed as a goal they want to reach, such as to achieve a grade in Taekwondo. Sometimes the children's desires are related to a person they look up to, such as their coach in a leisure activity.

Their desires for *significant others* are described as wanting an animal such as a horse or a dog since they have experienced the meaningfulness of being with animals in their riding school or at friends' homes.

'To be in horse stables, to go to riding camps and to swim with them, I want to be with horses all day long'

[b160]

The desire to have *significant skills* is described by the children as wanting a new skill. They describe how they want to be able to make pancakes, as in their everyday life situation with their family they help with preparing the batter. The children also express a desire to improve a skill that is hard to master by spending time practising. These skills include being able to swim above the water surface and ride a bike without support wheels. Other current skills they want to improve are using scissors, gymnastics or building with Kapla blocks.

To have *Significant activities* is also described by the children as a desire. This desire is both connected to the present time and space such as wanting to play football, climbing, have more gymnastics or to be with horses in the stables all day. However, the desire can also be related to a desire in the future or to significant others such as wanting to work with animals as a profession:

'When I and my little brother grow up, we will work on a farm with various animals'

[b30]

## Discussion

The results reveal that children born preterm are able to reflect and give detailed descriptions of situations of importance to them. The children are able to describe circumstances needed in their everyday life situations; they want these situations to occur more often, and they express ambitions to develop. The results indicate that from the perspective of preschool children born preterm, meaningful everyday life situations contain the significance of *doing things*, of *having significant circumstances* for that, and of *desires for further development*. For these children, meaningful everyday life situations can be understood as being in an active and dynamic process, which means that the children want to be active and do things, and that the circumstances

surrounding the being and the doing vary. Meaningful everyday life situations for the children in the study are the dynamic interaction between the doing, the significant others and significant places in the present time. Furthermore, meaningful everyday life situations also contain a past and a future represented by the circumstances and the desires. Doing things and being in an active process can be seen as expressions of the children as actors in their everyday life situations [2]. In their photographs and interviews the children share their proximal social environment, which provides opportunities and resources such as the school, the sports arena, and the stable. In the proximal social environment, the children experience and make meaning of the interactions among the contextual elements: people, places, activities, objects, and time [25]. It can be assumed that in this way the children are creators of their social context, and the interactions are therefore perceived as meaningful everyday life situations [25].

The results in the present study are in accordance with a previous photo-elicitation study where children with experiences of cancer treatment describe meaningful activities, relationships and environment as health promoting factors [26, 27]. In the previous study, the children with experiences of cancer treatment said their engagement in play and leisure activities such as jumping, dancing, or playing floor ball, and quieter activities such as listening to music and reading, gave them positive feelings and made them feel well [27].

The category *doing things* suggests that it is important for the children born preterm to be active and to be in an active context. *Doing things* is mainly about being physically active in different sports that are fun and may also be demanding. The children describe their skills in several activities, and they say that *having skills* allow them to engage in activities of everyday life they considered fun and important. Children have connected skilfulness with the ability to do a lot of things and with a feeling of wellbeing [5, 19]. Being skilled and doing things were considered expressions of health. The children born preterm in this study say that *doing things* also create a desire to improve their skills and to learn new ones. Studies report that typically developed children [5] as well as children with disabilities [28] consider activities fun and meaningful when the activities match their perceived abilities and needs even though the activities are hard to learn and require practice. In the present study the children express an ambition to learn and practice certain skills, especially as certain skills allow them to perform significant activities more often. The children say they want to do things in the near future e.g. join a soccer team but also further into the future e.g. when they become adults.

Although the children in the present study, say they like doing things on their own such as drawing and doing arts and crafts, they also appreciate *doing things with significant others* e.g., their siblings, parents, and playmates. The children with experiences of cancer treatment emphasised the relationships with others as meaningful to them. Doing things with significant others was considered fun but equally important was to have someone who was kind, contributed to a good mood and was there to share feelings [26]. The children born preterm reflect in a similar way, that *to have significant others* contribute to a feeling of being valued and loved and to feeling safe. Having significant others could also contribute to opportunities to experience special things, things that were exclusively connected to one significant other. For example, grandparents that have certain rules that e.g. allow the children to play computer games for a long period without interruptions or to eat sweets on days other than Saturdays. The children born preterm mention pets such as cats and dogs and horses as significant others to play with and to be close to. Caring for a pet has been reported as a most valuable activity [5] and children's fondness for pets has been related to the pets' contribution to feelings of companionship, closeness and comfort [27]. In the present study such pets could be those living in the children´s homes, pets that they sometimes borrow, or an expression of a desire to be with pets or other animals when they grow up. Pets as significant others could thus be found in all

three categories: as circumstances to feel loved, to do things i.e. play with, and to have a desire for development.

The results in our study connect to the fPRC model and may contribute to knowledge of children's participation in everyday life situations [15]. The children born preterm were asked to take photographs of meaningful everyday life situations, which can be interpreted as situations that hold meaning and thus are an expression of preferences in the fPRC model. Preferences for recreational activities such as playing board games or with a pet, have been related to young children [29]. In the interviews, the children describe the selected meaningful everyday life situations as often fun and important. *Having fun* has been described by children in previous studies as a factor contributing to a feeling of time passing quickly [5] and to meaningful participation experiences [30]. Children have also described feelings of competence in activities they perceive as fun and important and an increased willingness to improve their performance and skills in these activities [28], which is in accordance with the perceptions of the children born preterm in this study. The interest (preference) in an activity may develop further as the competence develops and the child learns to master the activity [31]. The study highlights the child's perspective of matters that are important and meaningful to them, matters which should be taken into consideration in health care when planning supportive and treatment interventions.

## Methodological considerations

There are certain strengths, as well as limitations, to consider that may affect the trustworthiness of the present study [32]. The children's active participation in all steps of the data collection procedure, i.e. taking pictures, choosing pictures for the interview and discussing topics of their own choosing, strengthens the trustworthiness of the study. The photo-elicitation interview is a method that may reduce the power imbalance between the researcher and the children [33]. The method may be particularly suitable for young children whose communication abilities are not fully developed [34] hence, ensuring the credibility of the data [32]. In the present study, the children were free to choose the theme for photography and to pick out the photographs they wanted to talk about, which meant that the interviewer (AKA) was not prepared for what the children would show or talk about. As the children were in need of support from their parents and teachers to take the photographs, there is a risk that the adults interfered in the choice of photographic objects and situations. The children were allowed to take as many photographs as they wanted and, prior to the interviews, select which ones to talk about. Also, during the interviews, the children decided which photographs to talk about and how much they wanted to say about their everyday life situations. The children's descriptions were encouraged by acknowledging their expertise regarding their own lives, and they expressed their happiness with the task. The triangulation of methods, photography and interviews, complemented each other and contributed to an increased understanding of the phenomenon, as it would have been difficult to understand the photographs without the children's descriptions. Also, it is probable that the photographs contributed to the children's descriptions being so rich and varied. The children's perspectives were further acknowledged in the data analysis and abstraction process by staying close to the descriptions the children provided, keeping the interpretation at a manifest level [32]. Incorporating triangulation in data collection and data analysis ensured that bias was reduced [35].

In some cases, the time between taking the photo and the interview exceeded one week, which may have affected the children's memory. On the other hand, the photographs served as reminders and the children were enthusiastic and spoke in detail about what the photographs represented. Considering the transferability of the results, the small number of children

participating is a limitation in this study. However, both girls and boys, with and without siblings, and from urban and rural areas were included. During the interviews, it was a challenge to be flexible and responsive to the way children describe a situation yet remain focused on the study aim. Moreover, there was a risk of over-interpreting the children's descriptions [23]. Therefore, the analysis procedure stayed close to the children's statements to maintain the children's perspective.

## Conclusions

The fact that young children born preterm can express and reflect on what is meaningful in their everyday life, could mean that they also can be considered competent in expressing their needs, desires, and ambitions for their participation in clinical practice. The study suggests that if six-year-old children born preterm are given the opportunity to share their views they can take an active part in planning and carrying through interventions in health care services.

## Supporting information

**S1 Table. Generic category, significant circumstances to do things.**
(PDF)

**S2 Table. Generic category, doing things.**
(PDF)

**S3 Table. Generic category, desire for significant development.**
(PDF)

## Acknowledgments

The authors wish to thank all the children and primary caregivers who contributed valuable information about their everyday life.

## Author Contributions

**Conceptualization:** Anna Karin Andersson, Lena Almqvist.

**Formal analysis:** Anna Karin Andersson, Maria Harder.

**Investigation:** Anna Karin Andersson.

**Project administration:** Anna Karin Andersson.

**Resources:** Katarina Strand Brodd.

**Supervision:** Lena Almqvist, Katarina Strand Brodd, Maria Harder.

**Validation:** Anna Karin Andersson, Maria Harder.

**Writing – original draft:** Anna Karin Andersson, Maria Harder.

**Writing – review & editing:** Anna Karin Andersson, Lena Almqvist, Maria Harder.

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
