## [Decision Letter · Decision Letter 0]

6 Sep 2022

PONE-D-22-19343Meaningful everyday life situations from the perspective of children born preterm: A photo-elicitation interview study with six-year-old childrenPLOS ONE

Dear Dr. Andersson,

Thank you for submitting your manuscript to PLOS ONE. After careful consideration, we feel that it has merit but does not fully meet PLOS ONE’s publication criteria as it currently stands. Therefore, we invite you to submit a revised version of the manuscript that addresses the points raised during the review process.

Please note that we have only been able to secure a single reviewer to assess your manuscript. We are issuing a decision on your manuscript at this point to prevent further delays in the evaluation of your manuscript. Please be aware that the editor who handles your revised manuscript might find it necessary to invite additional reviewers to assess this work once the revised manuscript is submitted. However, we will aim to proceed on the basis of this single review if possible. 

We look forward to receiving your revised manuscript.

Kind regards,

Steve Zimmerman, PhD

Associate Editor, PLOS ONE

Journal Requirements:

Reviewers' comments:

Reviewer's Responses to Questions

**Comments to the Author**

1. Is the manuscript technically sound, and do the data support the conclusions?

Reviewer #1: Yes

2. Has the statistical analysis been performed appropriately and rigorously? 

Reviewer #1: N/A

3. Have the authors made all data underlying the findings in their manuscript fully available?

Reviewer #1: No

4. Is the manuscript presented in an intelligible fashion and written in standard English?

Reviewer #1: Yes

5. Review Comments to the Author

Reviewer #1: The authors describe well why the child’s perspective is important, but do not specifically explain why exploring such perspectives in children born prematurely but not diagnosed with disabilities is important.

The section on ethical parameters should be located earlier in the methods section rather than its current location just before the results section. Putting the ethical section earlier informs the reader of Institutional Review Board approval prior to the recruitment of participants where ethical approval is particularly important.

The qualitative design is appropriate for this research. Photo elicitation is an important tool for qualitative research with children given the challenges describing their experiences. This is particularly true for children at age six where the childhood perspective and understanding is narrower than seen at an older age. Using photographs as a prompt provides appropriate support for efforts at eliciting descriptions.

The data analysis process is well described. However, what is not included is reference to trustworthiness of the data and triangulation of data collection/data analysis processes. These elements are critical in the discussion of qualitative research where study rigor is a concern.

The results section is well documented – the terms listed in the core category, generic categories and subcategories are logical and seem to be supported by the data as illustrated by quotes embedded in the text.

The discussion section expands on the categories identified in the results section and does so well. However, references to support findings are in large part those addressing activities and participation in children with disabilities. There are other existing studies addressing activities and participation in neurotypical children which should be woven into the discussion section. These concerns relate to the need to explain why children born prematurely were chosen for this study.

The theoretical perspective, Family of Person-related Constructs (fPRC), is mentioned in the discussion section. Using this model or others to guide the study would have generated a more meaningful background for the paper.

Rosenberg, L., Pade, M., Reizis, H., & Bar A.M. (2019). Associations between meaning of everyday

activities and participation among children. American Journal of Occupational Therapy, 73(6).

7306205030p1–7306205030p10. https://doi.org/10.5014/ajot.2019.032508

Shields, N., Adair, B., Wilson, P., Froude, E., & Imms, C. (2018). Characteristics influencing diversity of participation of children in activities outside school. American Journal of Occupational Therapy,

72, 7204205010. https://doi.org/10.5014/ajot.2018.026914

6. PLOS authors have the option to publish the peer review history of their article (what does this mean?). If published, this will include your full peer review and any attached files.

Reviewer #1: No

---

## [Author Response · Author response to Decision Letter 0]

22 Nov 2022

Thank you for valuable comments that will contribute the manuscript. The responds to the reviewer's and editor's comments are presented in a submitted rebuttal letter. They are also presented here. 

Journal requirements Revised version according to comments

Please ensure that your manuscript meets PLOS ONE's style requirements, including those for file naming The manuscript meets PLOOS ONE's style requirements, including those for file naming.

In your Data Availability statement, you have not specified where the minimal data set underlying the results described in your manuscript can be found. PLOS defines a study's minimal data set as the underlying data used to reach the conclusions drawn in the manuscript and any additional data required to replicate the reported study findings in their entirety. All PLOS journals require that the minimal data set be made fully available Respons: Due to the ethical approval, it is not possible to make the interviews fully available. However, we have added a table presenting an overview of the data analysis process from Meaning units to Core category and supporting information files to show the analysis process.

Please review your reference list to ensure that it is complete and correct. If you have cited papers that have been retracted, please include the rationale for doing so in the manuscript text or remove these references and replace them with relevant current references. Any changes to the reference list should be mentioned in the rebuttal letter that accompanies your revised manuscript. Respons: The reference list is reviewed and is complete and correct.

Reviewer’s comments Our answer

1. Is the manuscript technically sound, and do the data support the conclusions? Reviewer #1: Yes Respons: Thank you 

2. Has the statistical analysis been performed appropriately and rigorously? Reviewer #1: N/A

 -

3. Have the authors made all data underlying the findings in their manuscript fully available?Reviewer #1: No Respons: As the data consist of interviews with children we, for ethical reasons; preserve children’s confidentiality, make data fully available. 

4. Is the manuscript presented in an intelligible fashion and written in standard English? Reviewer #1: Yes Thank you

Reviewer #1. 

1. The authors describe well why the child’s perspective is important, but do not specifically explain why exploring such perspectives in children born prematurely but not diagnosed with disabilities is important. Respons: Thank you for your comment. In the Background section we have added arguments for why the perspective of children born preterm not diagnosed with disabilities is of interest. Being born preterm lead to more care and attention and the perception of the child as vulnerable may influence everyday life situations. 

2. The section on ethical parameters should be located earlier in the methods section rather than its current location just before the results section. Putting the ethical section earlier informs the reader of Institutional Review Board approval prior to the recruitment of participants where ethical approval is particularly important. Respons: The section Ethical Considerations has been put earlier in the method section.

3. The qualitative design is appropriate for this research. Photo elicitation is an important tool for qualitative research with children given the challenges describing their experiences. This is particularly true for children at age six where the childhood perspective and understanding is narrower than seen at an older age. Using photographs as a prompt provides appropriate support for efforts at eliciting descriptions. Respons: Thank you for this comment. We do very much agree with you. Using photo elicitation made it easier for the children to reflect on their everyday situations.

4. The data analysis process is well described. However, what is not included is reference to trustworthiness of the data and triangulation of data collection/data analysis processes. These elements are critical in the discussion of qualitative research where study rigor is a concern. Respons: Trustworthiness is now discussed in the methodological limitations of the study. 

5. The results section is well documented – the terms listed in the core category, generic categories and subcategories are logical and seem to be supported by the data as illustrated by quotes embedded in the text.

The discussion section expands on the categories identified in the results section and does so well. However, references to support findings are in large part those addressing activities and participation in children with disabilities. There are other existing studies addressing activities and participation in neurotypical children which should be woven into the discussion section. These concerns relate to the need to explain why children born prematurely were chosen for this study. Respons: Thank you for the suggestion to include studies on typically developed children. We have added studies and discuss our results in relation to studies on neurotypical children and children with disabilities. References have been added; Blencowe et al, 2012; Ballantyne et al, 2016; Elo et al, 2014; Rosenberg et al, 2019; Shields et al, 2018. 

6. The theoretical perspective, Family of Person-related Constructs (fPRC), is mentioned in the discussion section. Using this model or others to guide the study would have generated a more meaningful background for the paper. Respons: In the Background section we refer to the Bioecological system theory as a base for our understanding of children’s development. We relate to the fPRC also on the background section.

---

## [Decision Letter · Decision Letter 1]

2 Jan 2023

PONE-D-22-19343R1Meaningful everyday life situations from the perspective of children born preterm: A photo-elicitation interview study with six-year-old childrenPLOS ONE

Dear Dr. Andersson,

Thank you for submitting your manuscript to PLOS ONE. After careful consideration, we feel that it has merit but does not fully meet PLOS ONE’s publication criteria as it currently stands. Therefore, we invite you to submit a revised version of the manuscript that addresses the points raised during the review process.

We look forward to receiving your revised manuscript.

Kind regards,

Nabeel Al-Yateem, PhD

Academic Editor

PLOS ONE

Journal Requirements:

Reviewers' comments:

Reviewer's Responses to Questions

**Comments to the Author**

1. If the authors have adequately addressed your comments raised in a previous round of review and you feel that this manuscript is now acceptable for publication, you may indicate that here to bypass the “Comments to the Author” section, enter your conflict of interest statement in the “Confidential to Editor” section, and submit your "Accept" recommendation.

Reviewer #1: (No Response)

2. Is the manuscript technically sound, and do the data support the conclusions?

Reviewer #1: Yes

3. Has the statistical analysis been performed appropriately and rigorously? 

Reviewer #1: N/A

4. Have the authors made all data underlying the findings in their manuscript fully available?

Reviewer #1: No

5. Is the manuscript presented in an intelligible fashion and written in standard English?

Reviewer #1: Yes

6. Review Comments to the Author

Reviewer #1: The authors addressed the concern regarding the rationale for studying preterm children nicely. This addition has helped to understand the perspective guiding the study. Incorporating studies of typically developing children does provide a basis for comparison of the information generated from the children born prematurely.

The inclusion of the theoretical perspective, the Family of Person-related Constructs, has been added to the introduction as requested and has enhanced the text.

Moving the ethical approval section to earlier in the text supports that this study was reviewed by the appropriate authorities prior to the data collection.

The authors addressed triangulation and trustworthiness of the methods without using those terms. Given that this is a qualitative study, it is important to include the terms addressing study rigor. The photographs and interviews are two different methods of data collection, thus address triangulation. Using multiple data analysts is another method of researcher triangulation. The efforts made by the authors to ensure that the study participants were able to take, choose, and discuss topics of their own choice does address trustworthiness. It will make the study stronger to include those specific terms.

7. PLOS authors have the option to publish the peer review history of their article (what does this mean?). If published, this will include your full peer review and any attached files.

Reviewer #1: No

---

## [Author Response · Author response to Decision Letter 1]

15 Feb 2023

Thank you for your comments. The concepts of trustworthiness and triangulation is now added in Methods and Materials and in methodological considerations.

---

## [Decision Letter · Decision Letter 2]

27 Mar 2023

Meaningful everyday life situations from the perspective of children born preterm: A photo-elicitation interview study with six-year-old children

PONE-D-22-19343R2

Dear Dr. Andersson,

We’re pleased to inform you that your manuscript has been judged scientifically suitable for publication and will be formally accepted for publication once it meets all outstanding technical requirements.

Kind regards,

Nabeel Al-Yateem, PhD

Academic Editor

PLOS ONE

Additional Editor Comments (optional):

Reviewers' comments:

Reviewer's Responses to Questions

**Comments to the Author**

1. If the authors have adequately addressed your comments raised in a previous round of review and you feel that this manuscript is now acceptable for publication, you may indicate that here to bypass the “Comments to the Author” section, enter your conflict of interest statement in the “Confidential to Editor” section, and submit your "Accept" recommendation.

Reviewer #1: All comments have been addressed

Reviewer #2: All comments have been addressed

2. Is the manuscript technically sound, and do the data support the conclusions?

Reviewer #1: Yes

Reviewer #2: Yes

3. Has the statistical analysis been performed appropriately and rigorously? 

Reviewer #1: N/A

Reviewer #2: N/A

4. Have the authors made all data underlying the findings in their manuscript fully available?

Reviewer #1: No

Reviewer #2: Yes

5. Is the manuscript presented in an intelligible fashion and written in standard English?

Reviewer #1: Yes

Reviewer #2: Yes

6. Review Comments to the Author

Reviewer #1: The authors provide appropriate justification for interviewing children to learn about what they find enjoyable and meaningful based on both previous studies and a theoretical model, the Family of Person Related Constructs (fPRC). The rationale for addressing children born prematurely is logical.

The informed consent process for both children and parents is well-described. A qualitative research design was used in the study incorporating data collection via interviews. The photo-elicitation method was used to elicit those situations that were meaningful to them, as well as to facilitate their involvement in the interviews used in the research process. Details regarding the interview process used were provided.

The data analysis processes used and the resulting codes were clearly documented. A few quotes were woven into the results section. More quotes would have supported the codes in more depth, but this is not a pressing concern.

A previous concern was the lack of information regarding qualitative rigor processes. That concern has been addressed in this paper, where there is discussion of triangulation in the data analysis process and the data collection process. The use of the photo-elicitation procedure and the strengths and limitations that procedure were well documented and related to the qualitative research process.

The manuscript has been strengthened and in my opinion, is appropriate for publication.

Reviewer #2: Thank you for the opportunity to review this manuscript, which has already undergone two rounds of reviews. The authors have addressed the comments from prior reviews. I have a few suggestions, which the authors can consider.

1. Sharing examples of the exact wording of the questions posed to the children during the interview - this would be helpful to other researchers interested in conducting a photo-elicitation study with young children.

2. Methods - include details about the interviewer. What was their professional background? Prior experience with qualitative research and interviewing?

3. Ensuring the titles of the generic categories are consistent throughout. For example, on line 208 the title is "having significant circumstances" while the title used in Table 4 seems to be "Significant circumstances to do things".

4. Adding a few additional quotes to the manuscript. Some subcategories, but not all, have a sample quote provided in the manuscript. For example, there is only 1 quote provided for the generic category 'To have significant circumstances' (lines 225-226), but this category has 3 subcategories. For generic category 'Desire for significant development', there is a sample quote provided for 2/3 subcategories - could add a quote for 'To have desire for signficant skills'. The tables of supplementary quotes are a great addition, but a few additional quotes in the manuscript would benefit the readers who will not access the supplementary tables.

7. PLOS authors have the option to publish the peer review history of their article (what does this mean?). If published, this will include your full peer review and any attached files.

Reviewer #1: No

Reviewer #2: **Yes: **Kristin E. Musselman

---

## [Editor Report · Acceptance letter]

3 Apr 2023

PONE-D-22-19343R2 

Meaningful everyday life situations from the perspective of children born preterm: A photo-elicitation interview study with six-year-old children 

Dear Dr. Andersson:

I'm pleased to inform you that your manuscript has been deemed suitable for publication in PLOS ONE. Congratulations! Your manuscript is now with our production department. 

Kind regards, 

on behalf of

Dr. Nabeel Al-Yateem 

Academic Editor

PLOS ONE